# Plan and Then Act: The Moderated Moderation Effects of Profession Identity and Action Control for Students at Arts Universities during the Career Development Process

**DOI:** 10.3390/healthcare10101938

**Published:** 2022-10-03

**Authors:** Chia-Cheng Chen, Chao-Hsiang Hung

**Affiliations:** 1Teacher Education Center, National Taiwan University of Arts, New Taipei City 220, Taiwan; 2General Education Center, National Taiwan University of Arts, New Taipei City 220, Taiwan

**Keywords:** action control orientation, burnout, career self-efficacy, grit, profession identity

## Abstract

Preservice teachers at universities of arts have more than 10 years of professional training before admission, but in their senior year, they face the pressure of the graduation exhibition and performances and the teacher certification examination at the same time. This process is dissimilar to that for preservice teachers at general universities. Such a difference, however, has not been taken seriously in the past. In order to avoid burnout, preservice teachers at universities of arts, when they are under the pressure of limited time, may choose to identify with the departments they are more familiar with for their future careers, rather than identifying with their educational program, in order to increase hope for their career and reduce the chance of burnout. In addition, we believe that the use of action control/state control would also show different adaptation situations in the face of pressure. Therefore, this study focuses on the role of profession identity and action control as moderating variables in the process of becoming preservice teachers at arts universities. We recruited 304 art-major preservice teachers to establish a path model to explore their future time perspective and grit, detecting how the mediation of career decision self-efficacy affects learning burnout and career hope. Secondly, we inspected the moderating effect of profession identity and action control on learning burnout and career hope. We found that profession identity moderates the relationships between future time perspective and career decision self-efficacy as well as between career decision self-efficacy and learning burnout, all of which exhibited ordinal interactions. Furthermore, preservice teachers with high decision-making efficacy had lower burnout than those with low efficacy, but the high-efficacy advantage in preservice teachers under state control in reducing burnout would disappear. Lastly, although professional identification was important, action control regulated the relationship between career decision self-efficacy and learning burnout with ordinal interaction; that is, action control could effectively reduce their learning burnout.

## 1. Introduction

The Executive Yuan conducted an investigation on the employability of university students, asking new graduates whether they would choose to study in the same department if they could start over again. Of all respondents, 41.6% replied “no.” The main reasons for not choosing to study in the same department were “incompatible with interest” (42%) followed by “career after graduation was not as expected” [1]. The situation has not markedly changed after 15 years. According to the Statistics of Universities and Colleges published by the Department of Statistics, Ministry of Education in 2020, 21.3% of all students who dropped out of school attributed the action to their “learning interest”, and 25% of all the students who dropped out of school attributed the action to “incompatible with interest”. Interest was the main reason why university students quitted or dropped out of school, though this trend was not observed in art colleges.

In Taiwan, most university students begin to study majors in their fields after they enter university. However, students in universities of arts have professional experience of at least three years (starting from high school) and up to 12 years (with six years in elementary school, six years in junior high school, six years in high school plus four years in university) before enrollment. If they have no interest, they do not choose to enter universities of arts. However, many statistics show that arts graduates face a job market with “small salary, long hours, low security, and unstable work opportunities”.

Therefore, many university students of arts use future time perspective (hereafter referred to as FTP) [2] to foresee the future career barriers of art workers, and choose to pursue an educational program during their college years, looking forward to the opportunity to become an art teacher in primary and secondary schools after graduation.

However, being an art teacher in Taiwan means going through several challenges: they must first take the educational program selection exam, then take education credits before graduation, and lastly pass the teacher certification exam before they can participate in a half-year educational internship. After passing the teacher certification exam, they can obtain teacher’s certificates, and then take part in the few teacher selection exams under the impact of the low birthrate in order to have the opportunity to become a full-time teacher. Generally, university students complete most of their education credits in their senior year, and have more time to prepare for the teacher certification exam (held every June). For university students in arts, every department has graduation productions or graduation performances. These tasks usually take more than a year to accomplish. While students at other universities prepare for the teacher certification exam, preservice teachers at universities of arts must take care of their graduation exhibitions and performances. Many preservice teachers feel anxiety and even burnout [3].

FTP is a crucial aspect of the human cognitive system. It assists individuals to anticipate future challenges in their careers and prepare in advance. It also enables individuals to have higher self-efficacy and reduce learning burnout. FTP resembles a searchlight that illuminates the road in the distance: the stronger the searchlight is, the farther one can see, and the more clearly the target can be seen. Individuals use this to formulate plans and integrate all resources to achieve the target [2,4]. Nowack et al. [5] pointed out that people with stronger FTP have better behavioral performance. Adolescents with weaker FTP had higher frequency of drug and alcohol use and exhibited more high-risk behaviors. In addition, in an AIDS prevention survey, it was reported that female university students with stronger FTP had less risky sexual behaviors [6].

Peng [7] conducted a cross-lagged analysis with a sample of freshmen in Taiwan, and found that a student’s FTP affected his level of learning engagement in the next semester. Horstmanshof and Zimitat [8] studied Australian university students and found that the stronger the students’ FTP, the better their academic engagement, including adopting mastery goals, spending more time studying, seeking help for studies, etc. Additionally, Jung et al. [9] administered a survey to 98 Korean university students and found that FTP could positively and significantly predict students’ career decision-making self-efficacy (CDSE).

FTP can help individuals plan ahead for their future careers, while perseverance is an important trait to help individuals achieve their goals. In a longitudinal continuation of spelling bee studies, Duckworth et al. [10] indicated that grit was not only related to the duration of the subjects’ practice, but also was an important predictor of students’ spelling performance. In a related work, a positive association between grit and positive school outcomes was replicated in a sample of West Point cadets [11]. In addition, Ray and Brown [12,13] pinpointed that grit was more predictive of learning performance than learning strategies. Furthermore, Eskreis-Winkler et al. [14] studied high school students in Chicago, executing a longitudinal survey on whether they completed their studies, and observed that there was a significant positive correlation between grit and learning motivation among seniors in high schools. After controlling for situational factors, grit remained the most important predictor of whether students would graduate.

In career development-related research, Robertson-Kraft and Duckworth [15] studied teachers and found that if novice teachers were assigned to low-income or rural areas, they were more likely to quit than those assigned to other regions. Further analysis illustrated that a novice teacher’s grit could predict his teaching effectiveness and chance of staying on the job, which could not be predicted by teachers’ SAT scores or university GPAs, used as measures for assessment in the past. As well, Seguin [16] suggested that the grit of nurses affected their sense of well-being and fulfillment, and the higher the grit and the richer the work experience, the lower the burnout.

Studies related to educational psychology have indicated that self-efficacy is, among all relevant variables, the variable with the strongest predictability [17]. In line with Bandura, Lent et al. [18], proposed social cognitive career theory (SCCT), which states that the career choices of individuals are the results of the interactions between self-recognition and their environment or experiences. Career decision self-efficacy (CDSE) among individuals can be used to predict the future career interests and career choices of individuals and further affects their work performance. Furthermore, Xu et al. [19] found that student self-efficacy had significant positive correlations with well-being in a Taiwanese sample. Therefore, we proposed that the CDSE of preservice teachers can be used to predict their burnout and career hope. 

Preservice teachers at arts universities experience high levels of stress because of limited time and high academic pressure from their departments and their instructors in the educational programs. If they cannot manage this stress, they may suffer from burnout. Preservice teachers must make choices related to their profession identity and decide between their teacher identity (i.e., I am inclined toward becoming an art teacher in the future) and their department identity (i.e., I am inclined toward searching for jobs related to my department). In this study, we argued that the profession identity of preservice teachers of arts universities has a crucial moderation effect on their professional development. Studies reported that preservice teachers who adopted a teacher identity had superior career adaptability than those who adopted a department identity; that is, their career and identity choice led them to have a more satisfactory career path. Teacher identity is not necessarily superior to department identity, but if students in educational programs do not view themselves as becoming teachers in the future (teacher identity), cognitive and behavioral conflicts can ensue [3]. 

When individuals meet achievement challenges, the most direct solution is to actively change the current situation to complete the goal [20]. Kuhl [21] found that individuals exhibit two reactions when under pressure. Those with action control set up goals and take action while those with state control indulge themselves in the state of depression and lack concrete action to change the current situation. In this study, we hypothesized that action control or state control among preservice teachers had a moderation effect. The model proposed in this study included two moderating variables, namely, profession identity and action control. Profession identity moderated the relationship between grit, FTP, and self-efficacy, whereas action control moderated the relationship between profession identity, burnout, and career hope. This study is titled Plan and then Act. “Plan” refers to preservice teachers completing their career planning in relation to a primary goal of pursuing art professionally or becoming an art teacher (profession identity). “Act” refers to the action control required to achieve their goal. 

All in all, our study demonstrated that there were considerable differences between arts universities and ordinary university students before they enroll, but past research on arts universities was quite rare. Under the dual pressure of graduation exhibitions and performances and certification exams, the difference between preservice teachers’ professional identity and action control orientation could play an important role as a moderator.

The quality of teacher education represents the competitiveness of a country. Studies on teacher education have focused on the effect of the career choice motivation of preservice teachers on their future professional efficacy. These studies assumed that preservice teachers had a firm motivation to become teachers when they enrolled in teacher education programs, and ignored psychological mechanisms or events during teacher training and other factors influencing their career choices [22,23,24]. The first professional development stage for preservice teachers is the survival stage, which begins at the time preservice teachers enroll in their teacher education programs and ends when they become professional teachers [25]. Preservice teachers unable to adapt at this stage typically discontinue their studies. Although grit is a key quality for teachers, each country must establish a friendly and welcoming teacher education process to assist teacher education students. 

## 2. Materials and Methods

The career development model for preservice teachers at art colleges differs from that of preservice teachers at other universities. Therefore, we constructed a distinct model in this study, which is illustrated in Figure 1; the dotted lines in the figure represent the pathways between the moderating variables and their effects. This study investigated how the FTP and grit of preservice teachers of art colleges, as mediated by CDSE, affected burnout and career hope, and analyzed the moderation effects of profession identity and action control in preservice teachers at different stages. 

### 2.1. Study Design

The following research hypotheses were proposed based on the aforementioned literature review (Table 1).

### 2.2. Participants, Recruitment, and Data Collection

Using an online questionnaire, we enrolled 304 preservice teachers at an art college in Taiwan in 2022 as research subjects in this study. The valid response rate was 98.06%; 85.5% and 14.5% of respondents were female and male, respectively. 

### 2.3. Instrument Validity and Administration

FTP

This study referenced other relevant studies to develop an FTP scale [26,27]. The scale comprised two factors and ten items, specifically those related to career preparation (six items) and career cognition (four items), and their Cronbach’s α values were 0.84 and 0.81, respectively. Items were scored on a four-point Likert scale; a higher score indicated that the respondent had a higher FTP. 

2.Grit

This study referenced Duckworth et al. [28] to develop a grit scale comprising two factors and twelve items, specifically those related to passion (six items) and persistence (six items); their Cronbach’s α values were 0.86 and 0.85, respectively. Because the passion and persistence factors had a potential correlation of 0.93, this study defined grit as a psychological trait and viewed it as a single construct. Items were scored on a four-point Likert scale; a higher score indicated that the respondent had a higher level of passion or persistence. 

3.CDSE

This study referenced other relevant studies to develop a CDSE scale [29,30,31,32] comprising two factors and eight items, specifically those related to target selection (four items) and problem solving (four items); their Cronbach’s α values were 0.86 and 0.82, respectively. Items were scored on a four-point Likert scale; a higher score indicated that the respondent had higher CDSE. 

4.Burnout

We referenced Raedeke and Smith [33] and Gustafsson et al. [34] and considered the situation of preservice teachers when developing a burnout scale. The Cronbach’s α of the scale was 0.88. Items were scored using a four-point Likert scale; a higher score indicated that the respondent had a higher level of burnout. 

5.Career Hope

This study referenced Snyder [35] and Huang et al. [36] in developing a career hope scale. The scale consisted of two factors and thirteen items, specifically those related to goal path (eight items) and goal orientation (five items); the Cronbach’s α of the scale was 0.91. Items were scored on a four-point Likert scale, with a higher score indicating that the respondent had higher hope for their career development.

6.Profession Identity

This study referenced Lally and Kerr [37] and the expectancy–value theory of Wigfield and Eccles [38] in developing a profession identity scale. The scale used a single item assessing preservice teachers’ desire to become a teacher or to search for jobs related to their department in the future. The choices comprised “I only want to search for jobs related to my department = 1,” “both, but I am inclined toward searching for jobs related to my department = 2,” “both, but I am inclined toward becoming a teacher = 3,” and “I only want to become a teacher = 4.” Choices 1 and 2 were merged under department identity and coded as 0. Choices 3 and 4 were merged under teacher identity and coded as 1. Dummy variables were used as the moderating variables for the binary classification.

7.Action Control

We referenced the scale of Chen [39] and considered the situation of preservice teachers of art colleges when developing an action control scale. The scale had six items, and the Cronbach’s α was 0.75. An example item is as follows: “When I must complete a difficult task, I can focus on each part of the task/I easily lose focus on the task.” The action control scale used dichotomous variables; the action-oriented response was one, and the state-oriented response was zero. The total score of the scale ranged from 0 to 6; a higher score indicated that the respondent was more inclined to take action in learning. 

### 2.4. Data Analysis

This study used SPSS 21 and AMOS 21 to perform confirmatory factor analysis and used a path model to investigate mediation between variables. Next, this study employed PROCESS to investigate the moderation effect of profession identity and action control on the path coefficients [40]. To test the significance of the mediation and moderation effects, this study used a conventional *p*-value and bootstrap method. The parameter was significant if the 95% confidence interval (CI) did not include zero after 5000 samplings. The analysis of this study can be divided into two parts. The first part analyzed moderated mediation (MoMe). In terms of analysis procedure, the mediation effect was tested first, after which the significance of the moderation effect of profession identity was tested. MoMe is also known as the conditional indirect effect [41]. The second part assessed moderated moderation (MoMo). After the moderation effect of profession identity was validated, this study further investigated the effect of action control on the aforementioned moderation effect. In terms of analysis procedure, if the first-order interaction (predictor × profession identity) was significant, then the moderation effect was present. However, if the second-order interaction (predictor × profession identity × action control) was significant, then MoMo was present. 

## 3. Results and Discussion

### 3.1. Descriptive Statistics and Validity of Variables

This study had six continuous variables and one categorical variable (profession identity). The variables listed in Table 2 comprise FTP, grit, CDSE, and career hope. The average score for burnout, which has a negative connotation, was 2.2. The standard deviations of the variables ranged from 0.39 to 0.67, and the absolute values of their coefficients of skewness were all smaller than 1, indicating that all variables had normal distributions. In terms of profession identity, 133 respondents were inclined toward pursuing the profession associated with their department (43.7%), whereas 171 respondents were inclined toward becoming a teacher (56.3%). 

The validity of variables in the path model was tested using confirmatory factor analysis (CFA). As suggested by Byrne [42], Hu and Bentler [43], five fit indices were used to assess goodness of fit for CFA (GFI > 0.90, CFI > 0.90, TLI > 0.90, NFI > 0.90 and RMSEA < 0.08).

A test of CFA resulted in a relatively good fit to the data (χ2 = 61.97 ***, df = 34, GFI = 0.96, CFI = 0.98, TLI = 0.98, NFI = 0.97, RMSEA = 0.052). All standardized loadings were higher than 0.5 and statistically significant (*p* < 0.001). Composite reliability (CR) of variables ranged 0.73~0.87, average variance extracted (AVE) ranged 0.58~0.75, and both CR and AVE fit to the standard suggest by Fornell and Larcker [44] and Hair et al. [45]. There-fore, all of the latent variables appear to have been adequately operationalized by their respective indicators.

### 3.2. Path Model and Test of Mediation Effect

This study tested the mediation effect to understand whether the effect of FTP and grit on burnout and career hope was a direct effect or one mediated by CDSE. 

As presented in Table 3 and Figure 2, all direct effects were nonsignificant (*p* > 0.05). Therefore, the theoretical framework of this study was full mediation. 

As presented in Table 3 and Figure 2, the FTP and grit of preservice teachers of art colleges did not have a direct effect on their burnout or career hope. Instead, CDSE had a full mediation effect on the aforementioned effect. This result indicated that if preservice teachers had higher CDSE during their career development, CDSE could increase their career hope and reduce their burnout. Self-efficacy as a variable has the strongest predictability in educational psychology among all relevant variables [46]. Lent et al. [18] proposed SCCT, which included a few career development models, and self-efficacy was almost always a crucial primary determinant in these models. As illustrated in Figure 2, the regression coefficient of CDSE to career hope was 0.47, with CDSE representing the variable with the highest predictability. In this study, CDSE was measured through two factors, namely individuals clearly selecting their goals and solving their career problems. For teacher education institutions, assisting preservice teachers in understanding future career challenges and clarifying their career choices can strengthen preservice teachers’ competencies in facing challenges and increase their efficacy. 

### 3.3. Test of Moderation Effect

In terms of moderation effects, profession identity was the moderating variable when CDSE, burnout, and career hope were the criteria. This study further tested the effect of action control on the aforementioned moderation effects. This study used the regression-based PROCESS software as well as interactions to test whether the aforementioned moderation effects were significant. A moderation effect existed if the 95% CIs of the interactions listed in Table 4 did not include zero. 

This study further examined the first part of the null model, namely the moderation effect of profession identity on the paths FTP→CDSE and grit→CDSE. When FTP was the independent variable (predictor), $R^{2}$ was 41.4%, and the 95% CI was 0.009 to 0.530. Because the 95% CI did not include zero, the result was significant and indicated that profession identity could moderate FTP→CDSE. When grit was the predictor, $R^{2}$ was 40.7%, and the 95% CI was −0.010 to 0.539. Because the 95% CI included zero, the result was nonsignificant and profession identity could not moderate grit→CDSE. 

The second part of the null model centered on whether action control could moderate the moderation effect (MoMo) of profession identity on the path CDSE→burnout and the path CDSE→career hope. The testing of MoMo used a method similar to that of the triple interaction. However, moderation effects have directivity and, therefore, not all interactions were tested. The significance of the first-order interactions, which included profession identity, was used to test the moderation effect of profession identity. The significance of the second-order interactions, which included action control, was used to test the moderated moderation effect of action control. 

The analysis results revealed that when burnout was the criterion variable, $R^{2}$ was 23.4%, and the 95% CI of CDSE × profession identity was 0.082 to 0.968. Because the 95% CI did not include zero, the result was significant and indicated that profession identity indeed moderated the effect of CDSE→burnout. The moderated moderation effect of action control was further tested, and the 95% CI of CDSE × profession identity × action control was revealed to be −0.283 to −0.054. Because the 95% CI did not include zero, the result was significant and indicated that action control moderated the moderation effect of profession identity on CDSE→burnout. When career hope was the criterion, $R^{2}$ was 48.3%, and the 95% CI of CDSE × profession identity was −0.236 to 0.263. Because the 95% CI included zero, the result was nonsignificant and indicated that profession identity did not have a moderation effect on CDSE→career hope. The moderated moderation effect of action control on career hope was further tested, and the 95% CI of CDSE × profession identity × action control was −0.048 to 0.080. Because the 95% CI included zero, the result was nonsignificant and indicated that action control did not have a moderated moderation effect. 

Overall, the analysis results demonstrated that profession identity could moderate the paths FTP→CDSE and CDSE→burnout, and action control could further moderate the moderation effect of profession identity on burnout. In other words, FTP had to be combined with profession identity to have an effect on CDSE. When explaining burnout, both profession identity and action control must be considered. When grit was the predictor and career hope was the criterion, a direct effect was present, but a significant moderation effect was not. The research paradigm of SCCT indicated that when explaining positive variables such as career hope, efficacy had a moderate explained sum of squares. However, the situation became complicated when we attempted to explain negative variables such as burnout. As reported in Table 4, when predicting burnout, all interactions were significant except the interaction of CDSE × action control. This result highlighted the criticality of investigating the effect of profession identity and action control on burnout. The simple main effect was analyzed when two-factor interactions were significant; conditional moderation was investigated when triple interactions were significant, specifically to determine the special situations that caused the interactions. 

This study conducted visualization analyses for the different moderation effects to identify the mechanisms of the interactions. First, the overall moderation effect was analyzed, and profession identity was revealed to have had an effect during the early stage of the FTP→CDSE→burnout mediation model; profession identity could then be used to understand how CDSE was generated. During the later stage of the mediation model, the intervention of action control was required to effectively explain methods that reduced the burnout of preservice teachers, as depicted in Figure 3. 

The interaction diagram presents an analysis of how the profession identity of preservice teachers moderated the relationship between FTP and CDSE after the interactions were revealed to be significant. As illustrated in Figure 4, when CDSE was the criterion, the interaction between profession identity and FTP was an ordinal interaction. That is, regardless of their profession identity, preservice teachers with higher FTP had higher CDSE than those with lower FTP; however, due to the significant interaction effect, the effect of different FTP values on career decision self-efficacy (as represented on the *Y*-axis) was reinforced.

The other significant interaction was the relationship between profession identity and CDSE among preservice teachers. As depicted in Figure 5, profession identity and CDSE had an ordinal interaction. In other words, preservice teachers with higher CDSE experienced less burnout than preservice teachers with lower CDSE. However, we discovered that if profession identity was considered, teacher identity increased the discrepancy between the burnout of preservice teachers with high and low CDSE. As presented in Figure 5, unlike those with a teacher identity, preservice teachers with a department identity could avoid burnout due to playing multiple roles because they had a clear and specific direction for their career—inclining to be an art worker after graduation. Thus, as indicated in Figure 5, no matter whether they had high or low CDSE, preservice teachers with the two respective identities barely differed in burnout. However, preservice teachers with a teacher identity had to take care of the coursework from both their departments and their teacher education programs, since they wished to become art teachers in the future. Therefore, among those with a teacher identity, preservice teachers with lower CDSE evidently had a higher sense of burnout than those with higher CDSE. 

~department identity or teacher identity?

All research subjects were preservice teachers who had to excel in both their department and their teacher education program. As illustrated in Figure 4 and Figure 5, many of them acknowledged that they had limited time and resources, and chose between their department and teacher education programs to reduce their burden. If they were inclined toward becoming art teachers, then their learning during their teacher education program was consistent with the direction of their future career planning. However, if they identified themselves as taking up the work as professional art workers in the future and at the same time did not give up on their teacher education program, they could feel that they were wasting time due to the limited time resources. Under such circumstances, they might be prone to displaying maladaptive learning patterns (such as high learning burnout). Therefore, teacher education institutions should support preservice teachers in exploring their career planning, which is exactly what we refer to in English as “He knows what he wants” in the praise of someone [3].

### 3.4. Test of Moderated Moderation Effect

In this study, we followed the suggestions by Aiken and West [47] and used action control and state control groups to test the moderation effect of profession identity on CDSE and burnout. The results indicated that the moderation effect of action control was significant, as presented in Figure 6. The department identity and teacher identity groups were used to test the moderation effect of action control on CDSE and burnout. The results revealed that the moderation effect of teacher identity was significant, as depicted in Figure 7. As illustrated in Figure 6 and Figure 7, the second-order interactions explained the conditional moderation effect of action control in this study. 

Figure 6 depicts a disordinal interaction between profession identity and CDSE. Therefore, profession identity must be considered when evaluating the relationship between CDSE and burnout among preservice teachers. Preservice teachers with department identity and higher CDSE experience more burnout. Regardless of whether they adopt a teacher or department identity, preservice teachers must complete their teacher education program and undertake evaluations. If they wish to become art teachers in the future (teacher identity), then their current diligence would be necessary and worthwhile. However, some preservice teachers attend the teacher education program but wish to become art workers in fields related to their profession and department (department identity). They have limited time and resources but must direct both toward a professional field that they do not intend to pursue. Consequently, they had more burnout. Preservice teachers at art colleges must thoroughly assess their future career planning; otherwise, they are susceptible to experiencing burnout resulting from insufficient time and resources. 

~action control or state control

The initial analysis demonstrated that preservice teachers with a teacher identity had relatively less burnout than preservice teachers with a department identity. The following analysis investigated whether preservice teachers with a teacher identity had different experiences of burnout, particularly preservice teachers who took specific actions when working toward their goals (action control) compared to those likely to remain in an emotional state (state control). 

Figure 7 depicts the ordinal interaction between action control and CDSE. Thus, preservice teachers with higher CDSE had less burnout than preservice teachers with lower CDSE. However, if these preservice teachers had state control, their higher CDSE had a weaker effect on the reduction in burnout. 

The saying “Plan and then act” in Chinese means that an individual must first know the direction of future efforts and then put them into action before they can achieve their goals. “Plan” and “act” correspond to the two moderator variables in this study. “Plan” refers to a preservice teacher’s teacher identity or department identity, and “act” refers to action control [21]. We found that preservice teachers who had both action control and teacher identity had the lowest burnout (as shown in Figure 6), and preservice teachers who had action control could strengthen the resistance of career decision self-efficacy (CDSE) against burnout (such as Figure 7). Therefore, Kuhl [22] advocated that students should be taught volitional and self-regulatory strategies, which have a very positive effect on maintaining motivation and achieving goals.

## 4. Conclusions and Implications

International education trends have shifted from STEM to STEAM, placing more emphasis on the role of the arts. Gaining a solid foundation in the arts relies on comprehensive art teachers. However, becoming an art teacher in Taiwan is more difficult than becoming a STEM teacher. We cannot solely focus on career choice motivation among art college students who wish to become art teachers; we must also pay attention to the different challenges and mental adaptations of preservice art teachers during the teacher education process [23,24]. 

The conclusions and implications of this study, based on statistical analysis, are as follows:
FTP and grit among preservice teachers did not have a direct effect on their burnout and career hope; the effect was mediated by CDSE. Preservice teachers with higher CDSE had less burnout and higher career hope. This result showcases that career decision-making effectiveness (CDSE) is the key psychological mechanism that affects preservice teachers’ learning burnout and career hope. Either FTP or grit exercises its effect through CDSE. Teacher training institutions should strengthen preservice teachers’ sense of efficacy in making career decisions.Profession identity moderated the relationship between FTP and CDSE; FTP and CDSE had a significant ordinal interaction. Teacher identity increased the effect of FTP on CDSE. In other words, preservice teachers with a teacher identity and higher FTP had higher CDSE, whereas those with a teacher identity and lower FTP had lower CDSE. Although choosing to study in a teacher education program does not mean that they must be teachers in the future, we found that if preservice teachers could not decide whether they wanted to be teachers in the future, they would display a non-adaptive learning pattern. Therefore, we believe that career exploration is an urgent task for preservice teachers.In terms of MoMo, a moderated moderation effect of action control was observed. The CDSE of preservice teachers was more strongly moderated by their profession identity if they were more inclined toward action control. Preservice teachers with action control, higher CDSE, and a teacher identity had the least amount of burnout. This finding suggests that both profession identity and action control play a key role, as action control can effectively reduce burnout among preservice teachers regardless of their profession identity. Although preservice teachers at arts universities have less time resources and more pressure than those at general universities, if they can be taught strategies to take practical action, they can effectively reduce their burnout, resembling the well-known slogan of the famous sports brand Nike—“Just do it”— which is also why this study is entitled “Plan and Then Act”.

## Figures and Tables

**Figure 1 healthcare-10-01938-f001:**
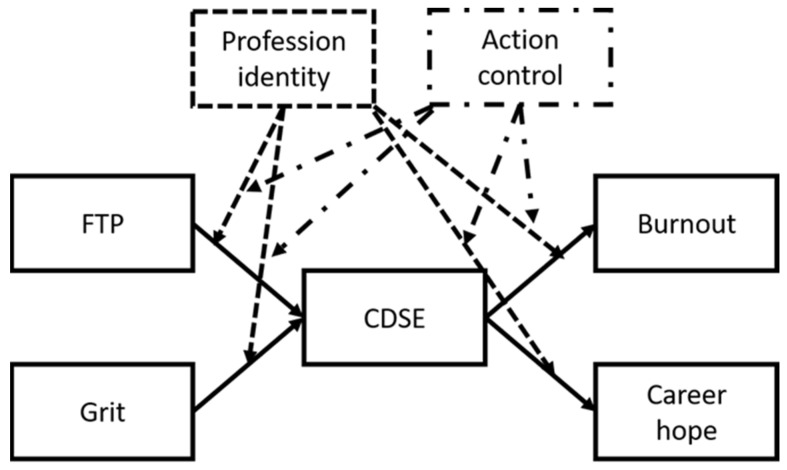
Study model.

**Figure 2 healthcare-10-01938-f002:**
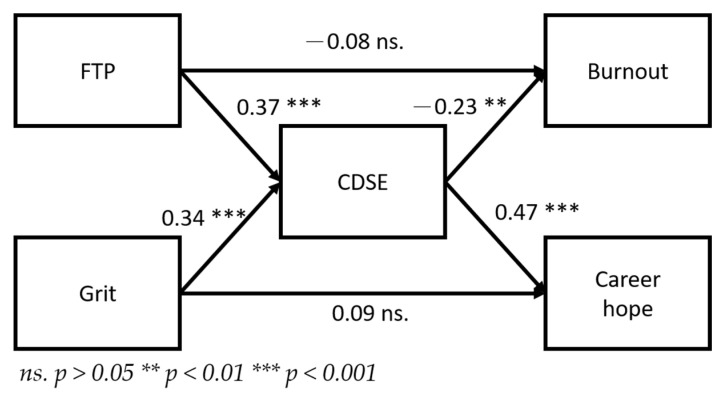
Path model with standardized solution.

**Figure 3 healthcare-10-01938-f003:**
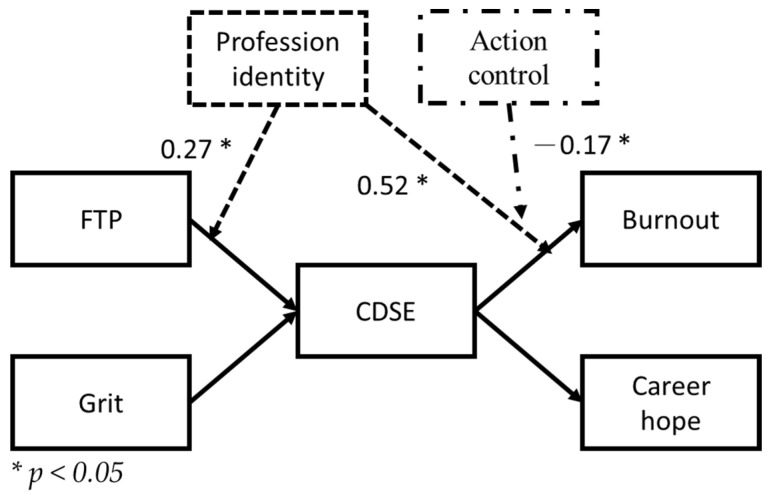
Path diagram of results of analysis of the moderation effects (only significant unstandardized solutions for the moderation effects are included).

**Figure 4 healthcare-10-01938-f004:**
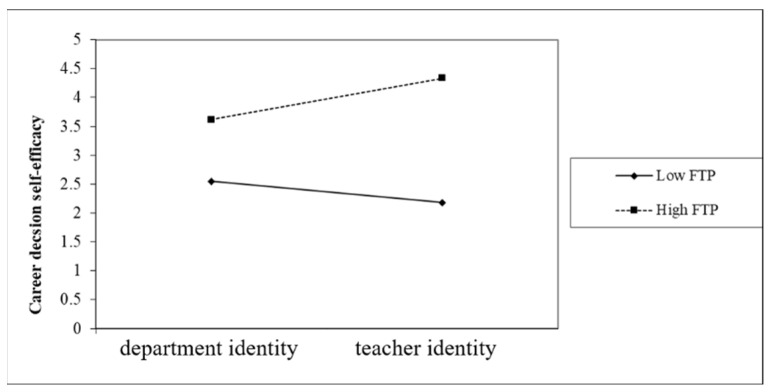
The moderation effect of profession identity on the relationship between FTP and CDSE.

**Figure 5 healthcare-10-01938-f005:**
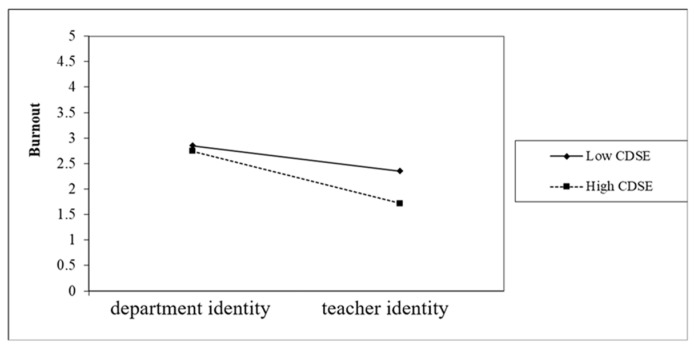
The moderation effect of profession identity on the relationship between CDSE and burnout.

**Figure 6 healthcare-10-01938-f006:**
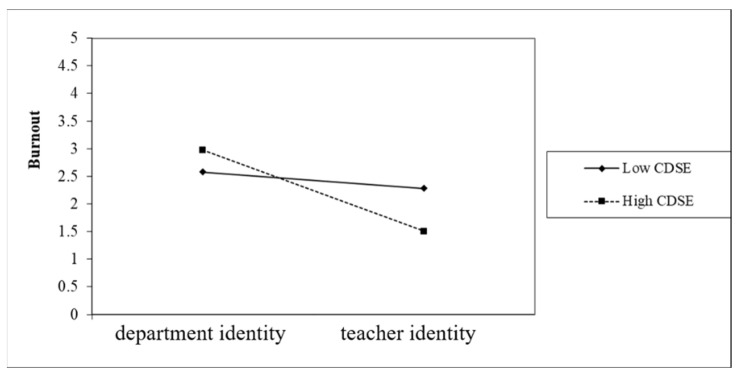
The moderation effect of the profession identity of preservice teachers with action control on CDSE and burnout.

**Figure 7 healthcare-10-01938-f007:**
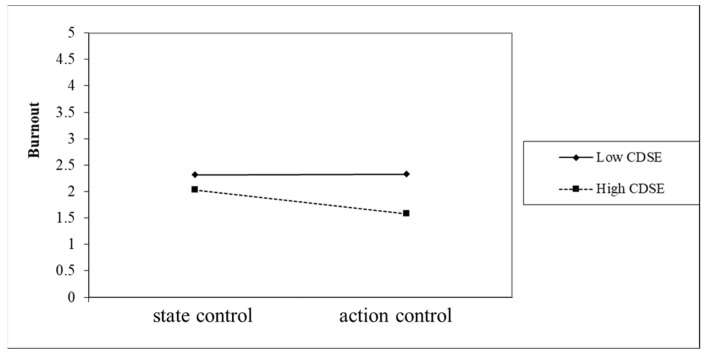
The moderation effect of the action control of preservice teachers with teacher identity on CDSE and burnout.

**Table 1 healthcare-10-01938-t001:** The research models and correspondence hypotheses.

Model	Hypothesis
Mediation	1. CDSE mediates the FTP–burnout relationship.
	2. CDSE mediates the FTP–career hope relationship.
	3. CDSE mediates the grit–burnout relationship.
	4. CDSE mediates the grit–career hope relationship.
Moderation	5. Profession identity moderates the FTP–CDSE relationship.
	6. Profession identity moderates the grit–CDSE relationship.
	7. Profession identity moderates the CDSE–burnout relationship.
	8. Profession identity moderates the CDSE–career hope relationship.
Moderated moderation	9. Action control moderates the moderation of profession identity on the FTP–CDSE relationship.
	10. Action control moderates the moderation of profession identity on the grit–CDSE relationship.
	11. Action control moderates the moderation of profession identity on the CDSE–burnout relationship.
	12. Action control moderates the moderation of profession identity on the CDSE–career hope relationship.

**Table 2 healthcare-10-01938-t002:** Descriptive statistics and correlation matrix.

Variables	Mean	SD	Skewness	1	2	3	4	5	6
1. FTP	3.62	0.41	−0.98	1					
2. Girt	3.58	0.39	−0.85	0.78	1				
3. CDSE	3.23	0.62	−0.54	0.63	0.63	1			
4. Burnout	2.20	0.67	0.12	−0.22	−0.27	−0.28	1		
5. Career hope	3.35	0.46	−0.18	0.55	0.58	0.65	−0.37	1	
6. Action control	3.71	1.89	−0.48	0.24	0.26	0.26	−0.29	0.40	1

All correlation coefficients were significant (*p* < 0.001).

**Table 3 healthcare-10-01938-t003:** Structure coefficient and mediation effect test.

Path	Standardized Structure Coefficient	Bias-Corrected Bootstrap 95% CI
FTP→CDSE	0.37 ***	
CDS→Burnout	−0.23 **	
FTP→Burnout	−0.08	
Grit→CDSE	0.34 ***	
CDSE→Career hope	0.47 ***	
Grit→Career hope	0.09	
FTP→CDSE→Burnout	−0.084 **	[−0.165, −0.038]
FTP→CDSE→Career hope	0.172 **	[0.099, 0.253]
Grit→CDSE→Burnout	−0.077 **	[−0.152, −0.029]
Grit→CDSE→Career hope	0.158 **	[0.089, 0.249]

** *p* < 0.01 *** *p* < 0.001.

**Table 4 healthcare-10-01938-t004:** Moderated-moderation regression coefficients and confidence intervals.

Criterion	Predictor	Unstandardized Regression Coefficient	Bootstrap 95% CI	
			LL	UL
CDSE $R^{2}=0.414$	constant	3.171 ns.		
	FTP	0.805 ***		
	profession identity	0.088 ns.		
	FTP×profession identity	0.270 *	0.0095	0.5300
CDSE $R^{2}=0.407$	constant	3.16 ns.		
	grit	0.836 ***		
	profession identity	0.105 ns		
	grit ×profession identity	0.265 ns.	−0.0101	0.5392
Burnout $R^{2}=0.234$	constant	3.750 ***		
	CDSE	−0.350 *		
	profession identity	−1.842 *		
	CDSE×profession identity	0.525 *	0.0821	0.9684
	action control	−0.228 ns.		
	CDSE×action control	0.053 ns.		
	profession identity×action control	0.497 **	0.1229	0.8721
	CDSE×profession identity×action control	−0.168 **	−0.2827	−0.0544
Career hope $R^{2}=0.483$	constant	1.843 ***		
	CDSE	0.383 ***		
	profession identity	0.042 ns.		
	CDSE×profession identity	0.013 ns	−0.2360	0.2627
	action control	0.053 ns.		
	CDSE×action control	0.0050 ns.		
	profession identity × action control	−0.069 ns.		
	CDSE×profession identity×action control	0.016 ns.	−0.0485	0.0799

ns. *p* > 0.05; * *p* < 0.05; ** *p* < 0.01; *** *p* < 0.001.

## Data Availability

Due to privacy and ethical concerns, neither the data nor the data sources can be made available.

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
