# Peer review of "Plan and Then Act: The Moderated Moderation Effects of Profession Identity and Action Control for Students at Arts Universities during the Career Development Process"

_healthcare, 2022, doi:10.3390/healthcare10101938_

Round 1
Reviewer 1 Report
1. In the abstract, the author fails to articulate a clear purpose and problem for the study.
2. When providing objective data, authors should update to the latest year.
3. In the Introduction, authors should cite more relevant literature.
4. In the Introduction, I would suggest that the author simplify and condense the content, which is too long and not focused enough.
5. The authors propose a research framework, but no corresponding hypotheses are made. It is recommended that the authors include hypothetical arguments.
6. The author may add to the theoretical discussion by referring to the following literature:
Peng, M. Y. P. (2022). Future Time Orientation and Learning Engagement Through the Lens of Self-Determination Theory for Freshman: Evidence From Cross-Lagged Analysis. Frontiers in Psychology, 5246.
Xu, P., Peng, M. Y. P., & Anser, M. K. (2021). Effective learning support towards sustainable student learning and well-being influenced by global pandemic of COVID-19: A comparison between mainland china and Taiwanese students. Frontiers in Psychology, 12, 561289.
Peng, M. Y. P., Feng, Y., Zhao, X., & Chong, W. (2021). Use of knowledge transfer theory to improve learning outcomes of cognitive and non-cognitive skills of university students: Evidence from Taiwan. Frontiers in Psychology, 12, 583722.
7. I don't see the author providing a CFA in the analysis, please can the author add to this section
8. In the conclusion, I do not see clear theoretical and practical implications.
Author Response
Responses to the comments of Reviewer #1
Point 1:In the abstract, the author fails to articulate a clear purpose and problem for the study.
Response: We appreciate the reviewer for the time and effort to review my manuscript. We agree with your suggestion and have elaborated on the purpose and problem in the revision.
Point 2:When providing objective data, authors should update to the latest year.
Response: We have specified the date for collecting the tata as below in lines 261 to 262. If, however, any missing information is missing on the latest dates, please inform us.
…Using an online questionnaire, we recruited 304 preservice teachers from an art university in Taiwan in 2022 as research subjects in this study.
Point 3:In the Introduction, authors should cite more relevant literature.
Response: We agree with you and have incorporated this suggestion throughout our introduction.
Point 4:In the Introduction, I would suggest that the author simplify and condense the content, which is too long and not focused enough.
Response: We agree with the reviewer and have modified and trimmed the introduction in a more refined manner.
Point 5:The authors propose a research framework, but no corresponding hypotheses are made. It is recommended that the authors include hypothetical arguments.
Response: We thank the reviewer for the advice. However, we believe that for the methodological consistence and the development of research rationale it is better for us to adopt the research-question format in this work. As a matter of fact, on a par with the hypothesis-construction format, the research-question format is used as the foundation for a scientific study, which pursues the issues in the question-answer configuration. Moreover, in line with the reviewer's comment, the introduction section has been properly modified and amplified in the revision in order to better capture the correspondence between our research framework and research questions.
Point 6:The author may add to the theoretical discussion by referring to the following literature:
Peng, M. Y. P. (2022). Future Time Orientation and Learning Engagement Through the Lens of Self-Determination Theory for Freshman: Evidence From Cross-Lagged Analysis. Frontiers in Psychology, 5246.
Xu, P., Peng, M. Y. P., & Anser, M. K. (2021). Effective learning support towards sustainable student learning and well-being influenced by global pandemic of COVID-19: A comparison between mainland china and Taiwanese students. Frontiers in Psychology, 12, 561289.
Peng, M. Y. P., Feng, Y., Zhao, X., & Chong, W. (2021). Use of knowledge transfer theory to improve learning outcomes of cognitive and non-cognitive skills of university students: Evidence from Taiwan. Frontiers in Psychology, 12, 583722.
Response: We thank the reviewer for the advice. We added two of above-mentioned articles in our theoretical discussion, which substantiates our findings.
Point 7:I don't see the author providing a CFA in the analysis, please can the author add to this section
Response: Thank you for your suggestion. We added the CFA between line 333 and line 352.
Point 8:In the conclusion, I do not see clear theoretical and practical implications.
Response: Response: We agree with your comments: the conclusion should be better focused on the theoretical and practical implications. Therefore, we have revised the conclusion as suggested.
Reviewer 2 Report
Thank you for the opportunity to review the manuscript. Overall, a current topic for a broader readership and further exploration of this topic is certainly interest, especially to explore the future time perspective and grid of university students of arts, detecting how the mediation of career decision self-efficacy affects learning burnout and career hope for university students in Taiwan!
A few questions / comments and suggestions:
In Line 31, the learning interest and relevance to the study is not clear.
In Line 110 to 112, relevance to the study is not clear.
In Line 136 to 138, relevance to the study is not clear.
In Line 335 to 338, relevance to the study is not clear, and please elaborate more detail.
In Line 346 to 350, relevance to the study is not clear.
In Line 357 to 359, "they had less interest in", suggest elaborating more detail with references.
In Line 403 to 409, "The saying "plan and then act" in Chinese", any evidences and relevance to the study is not clear.
In Line 424 to 446, the conclusions and implications of the study is ok, although it it not very clear how the authors moved from this to the themes presented.
Author Response
Responses to the comments of Reviewer #2
Point 1:In Line 31, the learning interest and relevance to the study is not clear.
Point 2:In Line 110 to 112, relevance to the study is not clear.
Point 3:In Line 136 to 138, relevance to the study is not clear.
Response: We agree with the three suggestions. They are constructive to polish our article, so we amended the introduction between line 50 and line 246 to render it clearer.
Point 4:In Line 335 to 338, relevance to the study is not clear, and please elaborate more detail.
Response: Thank you for your suggestion. The revisions are provided as below, and they are shown in the corresponding part with yellow highlights in the manuscript between line 447 and line 451.
That is, regardless of their profession identity, preservice teachers with higher FTP had higher CDSE than those with lower FTP; however, due to the significant interaction effect, the effect of distinct FTPs on career decision self-efficacy (as represented on the Y-axis) was reinforced.
Point 5:In Line 346 to 350, relevance to the study is not clear.
Response: Thank you for your suggestion. The revisions are presented as follows, and they are shown in the corresponding part with yellow highlights in the manuscript between line 460 and line 469.
As presented in Figure 5, unlike those with teacher identity, preservice teachers with department identity did not suffer from burnout due to playing multiple roles because they had a clear and specific direction for their career—inclining to be an art worker after graduation. Thus, as indicated in Figure 5, no matter whether they had high or low CDSE, preservice teachers with the two respective identities barely differed in burnout. However, preservice teachers with teacher identity had to take care of the course works from both their departments and their teacher education programs since they wished to become art teachers in the future. Such being the case, the preservice teachers with lower CDSE evidently had a higher sense of burnout than those with higher CDSE.
Point 6:In Line 357 to 359, "they had less interest in", suggest elaborating more detail with references.
Response: Thank you for your suggestion. The revisions are offered as below and they are shown in the relevant part with yellow highlights in the manuscript between line 481 and line 488.
However, if they identified themselves as taking up the work as professional art workers in the future and at the same time did not give up on their teacher education program, they might feel that they were wasting time due to the limited time resources. Under such circumstances, they might be prone to displaying maladaptive learning patterns (like high learning burnout). Therefore, teacher education institutions should support preservice teachers in exploring their careers planning, which is exactly what we refer to in English as “He knows what he wants” in the praise of someone [2].
Point 7:In Line 403 to 409, "The saying "plan and then act" in Chinese", any evidences and relevance to the study is not clear.
Response: Thank you for your suggestion. The modifications are provided as below, and they are shown in the corresponding part with yellow highlights in the manuscript between line 533 and line 542.
The saying "Plan and then act" in Chinese means that an individual must know the direction of future efforts first, and then put them into action before they can achieve their goals. “Plan” and “act” correspond to the two moderator variables in this study. “Plan” refers to preservice teacher’s teacher identity or department identity, and “act” refers to action control (Kuhl, 1987). We found that preservice teachers who had both action control and teacher identity had the lowest burnout (as shown in Figure 6); and preservice teachers who had action control could strengthen the resistance of career decision self-efficacy (CDSE) against burnout (such as Figure 7). Therefore, Kuhl (1985) advocated that students should be taught volitional and self-regulatory strategies, which have a very positive effect on maintaining motivation and achieving goals.
Point 8:In Line 424 to 446, the conclusions and implications of the study is ok, although it it not very clear how the authors moved from this to the themes presented.
Response: We agree with your advice. The conclusion should have been more focused to offer the theoretical and practical implications. We have therefore adopted this suggestion and rewritten the conclusion as described in that part (between line 566 and line 594).
Reviewer 3 Report
Congratulations to the Authors on a good idea.
Research is interesting and contributes to the development of science.
The research model is interesting and has practical application.
I have comments on the introduction part. I believe that the theoretical justification is too modest. The research review needs to be expanded.
Author Response
Point 1: Congratulations to the Authors on a good idea. Research is interesting and contributes to the development of science. The research model is interesting and has practical application. I have comments on the introduction part. I believe that the theoretical justification is too modest. The research review needs to be expanded.
Response:
Thank you for your praise and positive comments. I really appreciate your constructive suggestions and have rewritten the introduction section as the corresponding part in the article
Round 2
Reviewer 1 Report
Authors have paid a lots efforts to revise the paper. They have addressed my concerns. This paper can be accepted.